# Regulation of Smooth Muscle Cell Proliferation by Mitochondrial Ca2+ in Type 2 Diabetes

**DOI:** 10.3390/ijms241612897

**Published:** 2023-08-17

**Authors:** Olha M. Koval, Emily K. Nguyen, Dylan J. Mittauer, Karima Ait-Aissa, William C. Chinchankar, Isabella M. Grumbach

**Affiliations:** 1Abboud Cardiovascular Research Center, Division of Cardiovascular Medicine, Department of Internal Medicine, Carver College of Medicine, University of Iowa, Iowa City, IA 52242, USA; 2Fraternal Order of Eagles Diabetes Research Center, Division of Endocrinology and Metabolism, Carver College of Medicine, University of Iowa, Iowa City, IA 52242, USA; 3Free Radical and Radiation Biology Program, Department of Radiation Oncology, Holden Comprehensive Cancer Center, University of Iowa, Iowa City, IA 52242, USA; 4Veterans Affairs Healthcare System, Iowa City, IA 52246, USA

**Keywords:** calcium, proliferation, vascular smooth muscle cells, type 2 diabetes

## Abstract

Type 2 diabetes (T2D) is associated with increased risk of atherosclerotic vascular disease due to excessive vascular smooth muscle cell (VSMC) proliferation. Here, we investigated the role of mitochondrial dysfunction and Ca2+ levels in VSMC proliferation in T2D. VSMCs were isolated from normoglycemic and T2D-like mice induced by diet. The effects of mitochondrial Ca2+ uptake were studied using mice with selectively inhibited mitochondrial Ca2+/calmodulin-dependent kinase II (mtCaMKII) in VSMCs. Mitochondrial transition pore (mPTP) was blocked using ER-000444793. VSMCs from T2D compared to normoglycemic mice exhibited increased proliferation and baseline cytosolic Ca2+ levels ([Ca2+]cyto). T2D cells displayed lower endoplasmic reticulum Ca2+ levels, reduced mitochondrial Ca2+ entry, and increased Ca2+ leakage through the mPTP. Mitochondrial and cytosolic Ca2+ transients were diminished in T2D cells upon platelet-derived growth factor (PDGF) administration. Inhibiting mitochondrial Ca2+ uptake or the mPTP reduced VSMC proliferation in T2D, but had contrasting effects on [Ca2+]cyto. In T2D VSMCs, enhanced activation of Erk1/2 and its upstream regulators was observed, driven by elevated [Ca2+]cyto. Inhibiting mtCaMKII worsened the Ca2+ imbalance by blocking mitochondrial Ca2+ entry, leading to further increases in [Ca2+]cyto and Erk1/2 hyperactivation. Under these conditions, PDGF had no effect on VSMC proliferation. Inhibiting Ca2+-dependent signaling in the cytosol reduced excessive Erk1/2 activation and VSMC proliferation. Our findings suggest that altered Ca2+ handling drives enhanced VSMC proliferation in T2D, with mitochondrial dysfunction contributing to this process.

## 1. Introduction

Approximately 30% of individuals with T2D are affected by atherosclerosis and coronary artery disease [1]. Altered phenotypes of vascular smooth muscle cells (VSMCs) contribute to the excessive incidence of cardiovascular disease [2,3]. In physiological conditions, VSMCs reside in the tunica media of the arterial wall, regulating vascular tone through their contractile state [4]. However, when exposed to growth factors, VSMCs undergo a phenotypic switch, promoting proliferation and subsequent migration to the intima [5]. VSMC proliferation leads to build-up of atherosclerotic plaques and narrowing of arterial segments previously treated by balloon angioplasty [6].

VSMCs derived from patients with T2D or treated with insulin and glucose exhibit excessive proliferation [7,8]. Various cytosolic signaling pathways and downstream events have been implicated, including signaling via protein kinase C, the production of advanced glycation end products, the formation of reactive oxygen species (ROS) in response to ERK1/2/NFκB signaling, and the expression of PDK4 [6,9,10].

The role of cytosolic Ca2+ in regulating VSMC proliferation has received less attention [11,12]. Indeed, increased cytosolic Ca2+ levels ([Ca2+]cyto) and Ca2+ transients are mechanisms by which growth factors stimulate cell proliferation [13,14,15]. Importantly, [Ca2+]cyto is elevated in different cell types from T2D patients or diabetes mouse models, attributed to reduced Ca2+ export across the plasma membrane and impaired Ca2+ handling by the ER [16,17,18,19,20].

In VSMCs, [Ca2+]cyto is regulated by plasma membrane export and uptake into the ER and mitochondria [14,21,22]. Previous studies have shown that mitochondrial Ca2+ uptake is necessary for the proliferation [23,24] and migration [25] of VSMCs under normoglycemic conditions. 

However, T2D alters multiple aspects of mitochondrial function, including density, fission/fusion, respiration, and reactive oxygen species production [26]. It remains unclear whether mitochondrial Ca2+ handling is altered in VSMCs in T2D and contributes to increased proliferation [27].

This study aims to investigate whether VSMC proliferation in a diet-plus-low-dose streptozotocin-based mouse model of T2D is driven by increased [Ca2+]cyto, and the extent to which mitochondrial dysfunction contributes to this process. Specifically, we explore whether manipulating mitochondrial Ca2+ entry or extrusion alters VSMC proliferation in T2D. To block Ca2+ entry via the mitochondrial Ca2+ uniporter (MCU), we expressed the inhibitor peptide mtCaMKIIN, which inhibits Ca2+/calmodulin-dependent kinase II (CaMKII) selectively in mitochondria [25,28,29]. It is believed that mitochondrial CaMKII (mtCaMKII) phosphorylates the mitochondrial Ca2+ uniporter, enhancing Ca2+ entry into the inner mitochondrial matrix [25]. 

Lastly, we connect our findings to the activation of cytosolic growth pathways. Our data emphasize the significance of mitochondrial Ca2+ handling in cell proliferation in T2D and propose it as a potential strategy to combat neointimal hyperplasia in T2D patients.

## 2. Results

### 2.1. Enhanced Proliferation of VSMCs Isolated from T2D Mice Is Dependent on [Ca2+]cyto

To determine the dependence of VSMC proliferation on [Ca2+]cyto, we performed proliferation assays in aortic VSMCs from T2D and normoglycemic mice in the presence of the Ca2+ chelator BAPTA (1 µM). The addition of BAPTA eliminated the differences in cell proliferation between PDGF-treated VSMCs from T2D and normoglycemic mice (Figure 1A).

Next, we investigated the impact of inhibiting mitochondrial Ca2+ entry on VSMC proliferation. We cultured aortic VSMCs derived from T2D and normoglycemic mice and observed that VSMCs from T2D mice exhibited significantly increased proliferation, particularly in the presence of PDGF (Figure 1B).

We then examined the effect of inhibiting mitochondrial Ca2+ entry on proliferation. In these experiments, we used VSMCs from WT and transgenic mice expressing mtCaMKIIN in smooth muscle. Interestingly, the expression of mtCaMKIIN led to a significant reduction in VSMC proliferation in T2D mice, but not in normoglycemic mice (Figure 1B). Furthermore, we observed a strong inhibition of proliferation in VSMCs from T2D mice when mtCaMKII activity was blocked by overexpressing the inhibitor peptide mtCaMKIIN after adenoviral transduction in WT VSMCs (Figure 1C).

Preincubating VSMCs overexpressing mtCaMKIIN with the pharmacological MCU inhibitor Ru265 had no effect on cell proliferation. In WT VSMCs, it had a similar effect to mtCaMKIIN expression. These results support the notion that inhibiting mtCaMKII in mitochondria hampers proliferation by blocking the mitochondrial calcium uniporter (MCU) and, consequently, mitochondrial Ca2+ entry (Figure 1D,E).

### 2.2. Increased Cytosolic [Ca2+] in T2D Is Exacerbated by mtCaMKII Inhibition

Based on these findings, we investigated the impact of T2D on [Ca2+]cyto levels and cytosolic Ca2+ transients in response to PDGF stimulation. We observed that VSMCs from WT T2D mice displayed higher baseline [Ca2+]cyto compared to normoglycemic mice of the same background (Figure 2A,B). In preliminary studies, we confirmed that cytosolic Ca2+ sequestration into mitochondria is abolished when mtCaMKIIN is expressed in VSMCs (Appendix A). VSMCs from mtCaMKIIN T2D mice exhibited significantly higher [Ca2+]cyto compared to control WT T2D mice (Figure 2B).

We also recorded cytosolic Ca2+ transients triggered by PDGF in the presence of thapsigargin and observed attenuated responses in VSMCs from T2D mice (Figure 2C–E). Notably, VSMCs from mtCaMKIIN T2D mice showed the lowest transients in response to PDGF.

Furthermore, we examined the release of endoplasmic reticulum calcium ([Ca2+]ER) by thapsigargin and found that it was lower in T2D conditions compared to normoglycemic WT conditions (Figure 3A–C). Therefore, we investigated whether T2D affects [Ca2+]ER by altering the expression of endoplasmic reticulum and cytosolic Ca2+ handling proteins. Immunoblotting revealed a trend towards reduced expression levels of SERCA. Additionally, protein levels of IP3R were significantly increased with T2D, indicating enhanced Ca2+ loss from the endoplasmic reticulum in T2D. No difference in NCX expression was found between groups. (Figure 3D–G). The presence of mtCaMKIIN increased [Ca2+]ER levels and preserved SERCA and IP3R protein levels.

Next, we measured the baseline [Ca2+]mito, which was reduced in T2D and with mtCaMKII inhibition (Figure 4A). To study the effect of T2D on mitochondrial Ca2+ entry, we performed mtPericam imaging to measure mitochondrial matrix Ca2+ levels [Ca2+]mito in response to PDGF. We observed reduced mitochondrial Ca2+ entry in response to PDGF in WT VSMCs from T2D mice (Figure 4B–D). As expected, the presence of mtCaMKIIN further diminished the Ca2+ entry after PDGF treatment (Figure 4C,D).

We also assessed the mitochondrial membrane potential and found that VSMC mitochondria from T2D mice were more depolarized compared to those from normoglycemic mice, which provides a mechanistic explanation for the decrease in Ca2+ entry (Figure 4E). 

To eliminate the possibility that differences in the number of mitochondria per cell contribute to the variances in mitochondrial Ca2+ transients, we evaluated the ratios of mitochondrial to nuclear DNA using ND1/HK2 and Cox1/HK2. We found no significant differences between cells from T2D mice with and without mtCaMKIIN expression (Appendix A). In summary, in T2D, there are multiple factors that contribute to an elevation in baseline [Ca2+]cyto.

Decreased mitochondrial Ca2+ entry, potentially due to depolarization of the mitochondrial membrane potential, as well as ER dysfunction with decreased levels of SERCA and increased expression of IP3R favoring Ca2+ release, increase baseline [Ca2+]cyto. Inhibition of mitochondrial Ca2+ entry by mtCaMKIIN further exacerbates cytosolic Ca2+ overload and diminishes dynamic Ca2+ transients following PDGF stimulation.

### 2.3. Exaggerated Proliferation of T2D VSMCs T2D Is Driven by [Ca2+]cyto-Regulated MAP Kinase Activation and Blocked by mtCaMKIIN-Mediated Ca2+ Overload

To identify cytosolic signaling events responsible for driving cell proliferation in relation to [Ca2+]cyto, we conducted a screening of canonical growth pathways, including MAP kinases. However, we observed a significant increase in the phosphorylation of Erk1/2 in VSMCs from T2D mice compared to normoglycemic mice (Figure 5A). These findings are consistent with previous reports suggesting that cytosolic calcium levels [Ca2+]cyto can enhance Erk1/2 [30,31] phosphorylation. Additionally, we found that upstream regulators of the Erk1/2 signaling pathway, such as c-Raf and MEK, were activated in VSMCs isolated from T2D mice (Figure 5B). Furthermore, in T2D and NG conditions, the presence of mtCaMKIIN further intensified the activation of Erk1/2 and its upstream regulators (Figure 5A,B).

### 2.4. Erk1/2 Hyperphosphorylation in T2D Is Driven by Ca2+-Dependent CaMKII Activation in the Cytosol

We confirmed that Erk1/2 activation contributes to accelerated proliferation in T2D. Higher concentrations of the Erk1/2 inhibitor U0126 were required to reduce proliferation in T2D VSMCs compared to normoglycemic mice to levels below control conditions without PDGF (Figure 5C,D). To demonstrate that Erk1/2 activation directly depends on [Ca2+]cyto, we assessed Erk1/2 activation after pretreatment with the calcium chelator BAPTA (Figure 5E). The activation of Erk1/2 by PDGF was abolished by pretreatment with BAPTA. 

To investigate the involvement of cytosolic CaMKII, a Ca2+-dependent upstream regulator of Erk1/2, we examined its autophosphorylation at Thr287. As expected based on increased [Ca2+]cyto, we observed enhanced autophosphorylation of CaMKII at baseline in whole cell lysates of VSMCs from T2D mice (Figure 6A). The expression of mtCaMKIIN further augmented the activation of cytosolic CaMKII. To establish how CaMKII drives Erk1/2 activation, we tested whether CaMKII co-immunoprecipitates with c-Raf, the upstream regulator of Erk1/2 (Figure 6B). In T2D VSMCs, we observed increased association of c-Raf and CaMKII with PDGF treatment, implying activation of Erk1/2 by CaMKII with PDGF. Interestingly, while the expression of mtCaMKIIN enhanced the baseline association between c-Raf and CaMKII, the addition of PDGF did not further increase their association beyond baseline levels (Figure 6B). These data suggest that, in T2D VSMCs, PDGF drives Erk1/2 activation through CaMKII activation. With inhibition of mitochondrial Ca2+ entry through mtCaMKIIN leading to increased [Ca2+]cyto beyond the already elevated levels in T2D, the Erk1/2 pathway is activated. Under these conditions, the addition of PDGF (induces low cytosolic Ca2+ transient) does not further activate Erk1/2 signaling.

To examine the functional implications of cytosolic CaMKII, we inhibited its activity by adenovirus-mediated expression of the untargeted inhibitor peptide CaMKIIN in the cytosol. This intervention reduced the baseline activation of CaMKII, MEK, and Erk1/2, restored the activation of the signaling pathway by PDGF (Figure 6C), and decreased VSMC proliferation compared to the control group (Figure 6D).

### 2.5. Ca2+ Leakage through MPTP in T2D Contributes to Excessive VSMC Proliferation

We next examined the occurrence of transient Ca2+ leakage from mitochondria through the mitochondrial transition pore (mPTP) in T2D VSMCs, as previously reported in mitochondria from diabetic cardiomyocytes [32,33]. We conducted Ca2+ Green uptake assays and observed increased transition in T2D VSMCs compared to VSMCs from normoglycemic mice (Figure 7A).

To further investigate the impact of mPTP on [Ca2+]cyto, we measured [Ca2+]cyto after adding the mitochondrial uncoupler FCCP in the presence and absence of the mPTP inhibitor ER-000444793 (Figure 7B–E). We employed ER-000444793 because cyclosporin A, the commonly used Cyclophilin D-dependent mPTP inhibitor, also regulates proliferative pathways independently of mPTP by inhibiting calcineurin [34,35,36]. 

Pretreatment with ER-000444793 significantly increased Ca2+ release after FCCP, indicating the occurrence of mPTP leakage in T2D VSMCs. Moreover, treatment with ER-000444793 reduced baseline [Ca2+]cyto (Figure 7F). 

Lastly, we investigated the effect of mPTP opening on cell proliferation by conducting cell counts in the presence of ER-000444793 (Figure 7G). In T2D VSMCs, PDGF-induced cell proliferation was significantly reduced with ER-000444793 treatment, whereas this compound had no significant effect on cell counts in normoglycemic cells. These data suggest that Ca2+ leakage through mPTP in T2D contributes to elevated [Ca2+]cyto and promotes accelerated VSMC proliferation in T2D. 

## 3. Discussion

Our study aimed to identify molecular mechanisms underlying VSMC proliferation in type 2 diabetes (T2D). 

While growth factors promote cell proliferation by increasing [Ca2+]cyto and transients [13,14], the extent to which mitochondrial sequestering of Ca2+ is altered in T2D and drives VSMC proliferation has not been previously explored. In T2D patients and mouse models [16,37,38,39], [Ca2+]cyto is elevated as a result of impaired cytosolic Ca2+ handling [16,17,18,19,20]. 

Here, we propose that in T2D, altered mitochondrial Ca2+ handling increases [Ca2+]cyto and drives excessive VSMC proliferation. 

In support of this notion, we observed reduced Ca2+ entry into mitochondria and lower [Ca2+]mito that inversely correlated with high [Ca2+]cyto. We proposed a decrease in the mitochondrial membrane potential as a mechanism for decreased Ca2+ entry into mitochondria in T2D. Increased metabolic supplies with high glucose or fatty acid flux to mitochondria lead to the production of mitochondrial oxygen radicals that can lower the mitochondrial membrane potential [26]. Blocking the MCU with mtCaMKIIN further reduced Ca2+ entry and elevated [Ca2+]cyto, demonstrating that mitochondria serve as Ca2+ store in proliferating VSMCs. MCU inhibition increases [Ca2+]cyto under stress conditions in cardiomyocytes [40] and in non-excitable cell lines [21,41]. In our model, MCU inhibition further increases [Ca2+]cyto beyond what is seen in T2D alone.

Conversely, we saw evidence of leakage of Ca2+ through the mPTP, and blocking mPTP reduced both [Ca2+]cyto and VSMC proliferation in T2D. Thus, in T2D, both decreased Ca2+ entry into the mitochondrial matrix and leakage through mPTP increased [Ca2+]cyto.

The mPTP is a supramolecular complex at the interface of the inner and outer mitochondrial membranes [42]. Despite extensive research and several molecular candidates for the mPTP, its molecular nature remains contentious. Prolonged or permanent opening of the mPTP dissipates the membrane potential and inhibits ATP production [43]. However, it can also open and close transiently as a physiological Ca2+-efflux mechanism [44,45]. In the heart, mPTP opening has mostly been studied as permanent opening in the context of cell death in ischemia/reperfusion injury [43,45,46,47,48,49,50]. Cardiac mitochondria from diabetic patients show an increased sensitivity to Ca2+-induced mPTP opening compared with nondiabetic patients [32]. Similarly, mitochondria from cardiac myocytes in Zucker Fa/fa rats with type 2 diabetes were more sensitive to Ca2+ release from mPTP than mitochondria of control animals [33]. Our findings that leakage of Ca2+ via the mPTP pore increases [Ca2+]cyto and promotes VSMC proliferation provide, to our knowledge, the first evidence for Ca2+ extrusion from mitochondria as a driver of cell proliferation. 

Increased [Ca2+]cyto affects numerous functions in VSMCs, including contraction and vascular tone, and regulates cell cycle progression through activation of cytosolic CaMKII. While high cytosolic CaMKII activity has been reported in cardiac myocytes under hyperglycemia and various diabetes models [51,52,53], our study adds to this concept by defining its role in cell proliferation. One of the proposed mechanisms through which CaMKII drives proliferation is through Erk1/2 [54,55,56,57]. We demonstrate that Erk1/2 and CaMKII associate with c-Raf to drive VSMC proliferation [58]. Whereas T2D alone activates Erk1/2 signaling, we posit that MCU inhibition in T2D hyperactivates Erk1/2 signaling to the extent that treatment with a growth factor like PDGF has no additional effect on Erk1/2 activation, effectively inhibiting proliferation.

Our study has several limitations. While we concentrate on how mitochondrial dysfunction in T2D alters [Ca2+]cyto and on cytosolic signaling pathways, mitochondria also modulate VSMC proliferation through the production of ATP, precursors of macromolecules and intracellular redox homeostasis as well as enhanced glycolysis and fatty acid metabolism. Reprogramming of glycolytic pathways and lipid metabolism have recently been identified as drivers of phenotypic switching of VSMCs from a differentiated contractile to a de-differentiated proliferative VSMC and neointima formation [59]. Moreover, prior studies have defined a role of cytosolic Ca2+ handling in VSMC proliferation and neointima formation [60,61]. Investigating the contribution of mitochondrial metabolic activity, such as Ca2+-dependent ATP production or intermediates of the TCA cycle, to VSMC dedifferentiation and enhanced VSMC proliferation in T2D, would be important. Ongoing studies are currently exploring these aspects.

Studies on VSMC proliferation as a driver of neointimal hyperplasia after balloon angioplasty have mainly focused on normoglycemic conditions [25,62,63,64]. However, considering that about 40% of patients undergoing balloon angioplasty have T2D, and many others meet the criteria for metabolic syndrome [65], it is unclear whether the reported findings are directly relevant to patients and helpful in developing new therapies. Our study emphasizes the importance of mitochondrial Ca2+ handling in cell proliferation in T2D, suggesting that it as a potential strategy to combat neointimal hyperplasia. It also supports the notion that altered intracellular Ca2+ metabolism may represent a common underlying abnormality linking the metabolic and cardiovascular manifestations of the diabetic disease process, as proposed by Levy and colleagues in the past [39].

## 4. Materials and Methods

### 4.1. Genetic Mouse Models

All animal procedures were approved by the University of Iowa Institutional Animal Care and Use Committee. This study was carried out in strict accordance with the recommendations in the Guide for the Care and Use of Laboratory Animals of the National Institutes of Health (NIH).

Studies conducted with protocol number: 0051189 and Approval date: 7 June 2020.

Mice of the C57BL/6 background that express tamoxifen-inducible Cre recombinase (driven by the smooth muscle myosin heavy chain, SMMHC, promoter) were obtained from Jackson Laboratories (#019079, denoted as “SM-Cre” mice). HA-tagged mtCaMKIIN mice were generated by cloning a cDNA that encodes an HA-tagged form of the CaMKII inhibitor peptide CaMKIIN (HA-CaMKIIN) fused to the mitochondria-targeting Cox8-palmitoylation sequence into a construct that contains the CX-1 promoter and a floxed eGFP sequence. Double-transgenic mice expressing mtCaMKIIN specifically in VSMCs (denoted as “mtCaMKIIN mice”) were generated by crossing SM-CreERT2 mice with HA-tagged mtCaMKIIN mice [29]. Eight-week-old male SM-mtCaMKIIN mice were treated with a 5-day course of tamoxifen (20 mg/mL, i.p. injection, 5 days) to induce Cre recombination, and this regimen was repeated starting 15 days after the beginning of the first tamoxifen course. 

SM-Cre mice treated with tamoxifen were used as controls for SM-mtCaMKIIN mice. Correct recombination and mtCaMKIIN expression were previously established (Nguyen et al. [25]). Because the SM-Cre transgene is on the Y-chromosome, only male mice were studied.

### 4.2. Diet-Induced Type 2 Diabetes

At 8–12 weeks of age and SM-mtCaMKIIN mice and littermate control mice were fed a high-fat diet (HFD, 60% fat). After 8 weeks, streptozotocin (STZ) (75 mg/kg, 50 mg/kg) was administered by i.p. injections as two separate doses, per published protocols [64,65]. Normoglycemic control mice were kept on normal chow. At 12 weeks after initiation of the high-fat diet, we confirmed that this model recapitulates T2D phenotypes, including increased body weight up to 40 g, random blood glucose levels above 250 mg/dL, impaired glucose tolerance, hypercholesteremia, and hyperinsulinemia. No significant differences were observed between the genotypes.

### 4.3. Cell Culture

Primary aortic VSMCs were isolated from mtCaMKIIN, and control mice by enzymatic digestion. The aortas were incubated in elastase (to remove adventitia) and cut into rings (about 1 mm in length) that were subsequently incubated in 2 mg/mL collagenase type II (Worthington Biochemical Corporation, Lakewood, NJ, USA) for 2 h. The digested pieces were plated in DMEM with 450 mg/dL glucose, 1% penicillin/streptomycin, MEM nonessential amino acids, MEM Vitamin and 8 mmol/L HEPES) supplemented with 20% fetal bovine serum (FBS) and 0.1% fungizone. The cells were cultured in DMEM with 10% FBS at 37 °C in a humidified incubator (95% air and 5% CO_2_). Cells were routinely tested for mycoplasma contamination.

### 4.4. Adenoviral Transduction

Adenoviruses expressing mtCaMKIIN (Ad-mtCaMKIIN), untargeted CaMKIIN (Ad-CaMKIIN), or mito-Pericam (Ad-mtPericam), as well as empty vector (Ad-control), were generated by the Viral Vector Core Facility at the University of Iowa. 

### 4.5. Measurement of Basal Mitochondrial Calcium

Mitochondrial calcium content was measured in isolated mitochondria using a Cayman Chemical Calcium Assay kit (Item No. 701220). Readings were normalized to protein concentration.

### 4.6. Measurement of Cytosolic Ca2+ Transients

Cells were loaded with Fura-2 acetoxymethyl ester (Fura-2AM) by incubation with 2 µM Fura-2AM in HBSS for 20 min at room temperature, then washed twice and incubated at 37 °C for 5 min for de-esterification. Cells were excited alternatively at 340 and 380 nm. Intensity of the fluorescence signal was acquired at 510 nm. Real-time shifts in the Fura-2AM fluorescence ratio were recorded before, during, and after acute addition of PDGF (20 ng/mL) using a Nikon Eclipse Ti2 inverted light microscope. Peak amplitude was calculated by subtracting the baseline fluorescence ratio from the peak fluorescence ratio. The area under the curve (AUC) was determined using GraphPad Prism and normalized by subtracting the AUC at baseline. Summary data represent the average differences in the basal and peak increases in [Ca2+]cyto.

The free [Ca2+]cyto was calculated from Fura-2AM fluorescence using the equation [Ca2+]cyto = (R − Rmin)/(Rmax − R) · Kd · β, where R is the basal ratio of fluorescence at 340 nm to 380 nm, Kd is the Fura-2AM dissociation constant (145 nM), and β is the difference in Fura fluorescence intensity in Ca2+-free vs. Ca2+-saturated media.

### 4.7. Measurement of Mitochondrial Ca2+ Transients

Ratiometric Ca2+ measurements in mitochondria were performed using adenovirus-delivered mtPericam (Ad-mtPericam), a fluorescent Ca2+ indicator protein with a COX IIIV targeting sequence. VSMCs were infected with Ad-mtPericam 48 h prior to analysis. Ratiometric fluorescent imaging of mtPericam via a Nikon Eclipse Ti2 microscope was used to determine the intensity of the fluorescence signal, with excitation at 415 nm and 480 nm, with emission at 510 nm. Peak amplitude and AUC were determined as described for Fura-2 AM.

### 4.8. Measurement of ER Ca2+

Cepia ER plasmid was purchased from Addgene (#58215). The CEPIA1er protein was excited at 543 nm, and emitted fluorescence was measured at wavelengths of 580 nm. 

VSMCs were transfected in a Nucleofector I device (Lonza, Basel, Switzerland) using the Basic Nucleofector Kit for Primary Mammalian Smooth Muscle Cells (VPI-1004, Lonza) and following the manufacturer’s protocol. 50,000 cells were transfected in the presence of 1 μg of plasmid DNA, plated onto 35-mm glass bottom microwell dishes (MatTek Corporation), and grown for 48 h before experiments were performed.

### 4.9. Ca2+ Retention Assay

Calcium Green-5N (or Fura-2) was used to monitor cytosolic Ca2+ in permeabilized VSMCs. Signal was recorded in a 96-well plate. Total volume of reaction was 100 µL with 50 µL of cell suspension (2,000,000 per well) and 50 μL of respiration buffer (100 mM K aspartate, 20 mM KCl, 10 mM Hepes, 5 mM glutamate, 5 mM malate, and 5 mM succinate (pH 7.3)) supplemented with 5 μM thapsigargin, 0.005% digitonin, and 1 μM Calcium Green-5N (or Fura-2) (Invitrogen, Waltham, MA, USA). Fluorescence was monitored at 485 nm excitation and at 535 nm emission. Following baseline measurement, CaCl2 (10 μM Ca2+ at 30 °C) was added sequentially every 4 min for 5 times until Ca2+ uptake ceased.

### 4.10. qPCR to Determine Mitochondrial DNA Copy Number

Genomic DNA was isolated from approximately 500,000 VSMCs using the DNeasy Blood & Tissue extraction kit (Qiaqen, Hilden, Germany). DNA samples were treated with RNase and subsequently quantified using the Power SYBR Green Master Mix real-time PCR kit, with 100 ng genomic DNA per reaction (Thermo Fisher Scientific, Waltham, MA, USA, #4367659). Mitochondrial DNA copy number was calculated using the 2−ΔCt formula, where Δ = Ctmito − Ctnuclear, for selected mitochondrial (ND1, COX1) and nuclear (HK2, NDUFV1) genes.

### 4.11. Measurement of Mitochondrial Membrane Potential

The mitochondrial membrane potential was measured using the tetramethylrhodamine methyl ester (TMRM, 50 nM, Life technologies, Carlsbad, CA, USA, T668) and Mitotracker Green (100 nM, Thermo Fisher Scientific, Waltham, MA, USA, M7514) fluorescent probes according to the manufacturers’ protocols. Images were taken at baseline and 15 min after FCCP treatment (5 μM, Sigma, St. Louis, MO, USA, C2920), using an LSM 510 confocal microscope at a magnification of 40× (Carl Zeiss, San Diego, CA, USA). Image analysis was performed using NIH ImageJ. All images were taken at the same time and using the same imaging settings. Data are presented as fold change over control with reference to the completely depolarized state.

### 4.12. Cell Lysis and Fractionation

Whole cells were lysed in RIPA buffer (20 mM Tris, 150 mM NaCl, 5 mM EDTA, 5 mM EGTA, 1% Triton X-100, 0.5% deoxycholate, 0.1% SDS, pH 7.4) supplemented with both protease inhibitors (Mini complete, Roche, Basel, Switzerland) and phosphatase inhibitors (PhosSTOP, Roche). Lysates were sonicated and debris was pelleted by centrifugation at 10,000× *g* for 10 min at 4 °C. Mitochondrial fractions were prepared in MSEM buffer (5 mM MOPS, 70 mM sucrose, 2 mM EGTA, 220 mM Mannitol, pH 7.5 with protease inhibitors), with homogenization performed in cold MSEM buffer in a Potter-Elvehjem glass Teflon homogenizer (50 strokes). Nuclei and cell debris were pelleted by centrifugation at 600× *g* for 5 min at 4 °C. Mitochondria were separated from the cytosolic fraction by centrifugation at 8000× *g* for 10 min at 4 °C. Protein concentrations were determined using the Pierce™ BCA protein assay (Thermo Scientific).

### 4.13. Immunoblotting and Immunoprecipitation

For immunoprecipitation assay, cells were lysed in Pierce IP Lysis Buffer (Thermo Scientific, 87788). For each condition, 300 µg of proteins were used and incubated with primary antibody overnight at 4 °C degrees, and then with magnetic Dynabeads Protein G (Invitrogen, 58096) for 1 h at room temperature. Dynabeads with precipitated proteins were washed 3 times with DynaMag-2 (Invitrogen) and eluted with LDS buffer (Invitrogen, 58011).

Equivalent amounts of 5–15 μg of protein for gel loading (cell lysates, mitochondrial/cytoplasmic fractions) were separated by SDS/PAGE on 4–20% Tris/glycine precast gels (Bio-Rad, Hercules, CA, USA) and transferred to PVDF membranes (BioRad). Membranes were blocked in 5% BSA and incubated overnight at 4 °C with primary antibodies. Blots were washed 3 times for 10 min with 0.05% Tween-20 in TBS, incubated for 1 h at room temperature with the respective secondary antibodies, and then washed again. The blots were then developed using ECL chemiluminescent substrate (Thermo Scientific, 34580) according to the manufacturer’s instructions.

### 4.14. Proliferation Assays

VSMCs were cultured in 12-well plates at 5000 cells per well in DMEM containing 10% FBS. At 24 h after plating they were treated with PDGF (20 ng/mL), and at 72 h after they were trypsinized and triplicate samples were counted using a Beckman Coulter Z1 cell counter.

### 4.15. MTT Assay

VSMCs were plated in a 96-well plate with a density of 1000 cells/well in 72 h before performing the MTT assay. After attaching, cells underwent PDGF treatment (20 ng/mL) and BAPTA treatment (1 µM) for 48 h. MTT stock solution was made in PBS at 5 mg/mL. 10 µL of MTT stock solution was added into each well with 90 µL of media in. After 2 h of incubation media was aspirated, cells washed with PBS once and intracellular MTT formazan crystals were dissolved in 50 µL of DMSO. Color intensity (absorbance) was measured on a microtiter plate reader at 570 nm light wavelength. 

### 4.16. Statistical Analyses

Data are expressed as mean ± SEM and were analyzed using the GraphPad Prism 9.0 software. All data sets were analyzed for normality and equal variance. The Kruskal–Wallis test with Dunn’s post hoc test was used to assess data sets where normal distribution could not be assumed. Student’s T-test and one-way ANOVA followed by Tukey’s multiple comparison test were used for data sets with normal distributions. Two-way ANOVA followed by Tukey’s multiple comparison test was used for grouped data sets. A *p*-value of <0.05 was considered significant.

## 5. Conclusions

Here, we present several key findings: VSMCs from T2D mice exhibited excessive proliferation and elevated [Ca2+]cyto. T2D decreased mitochondrial Ca2+ transients in response to PDGF administration compared to normoglycemic conditions. Moreover, we observed Ca2+ leakage through the mPTP as well as for ER Ca2+ depletion. Activation of Erk1/2 and its upstream regulators was enhanced in VSMCs from T2D mice, driven by increased [Ca2+]cyto. Inhibition of mitochondrial Ca2+ entry through mtCaMKIIN increased [Ca2+]cyto, leading to exaggerated Ca2+ imbalance and baseline Erk1/2 hyperactivation. This inhibited further PDGF-induced Erk1/2 activation and cell proliferation. Inhibition of cytosolic Ca2+-dependent signaling reduced excessive Erk1/2 activation and decreased VSMC proliferation. Furthermore, blocking mPTP reduced [Ca2+]cyto and proliferation in VSMCs from T2D compared to normoglycemic mice. Thus, in T2D, a condition with elevated [Ca2+]cyto, both lowering of [Ca2+]cyto by mPTP inhibition as well as further increasing it by MCU inhibition can reduce cell proliferation, suggesting a “sweet spot” or “goldilocks effect” for [Ca2+]cyto to promote proliferation. These findings provide novel insights into the role of altered Ca2+ handling and mitochondrial dysfunction in VSMC proliferation in T2D.

## Figures and Tables

**Figure 1 ijms-24-12897-f001:**
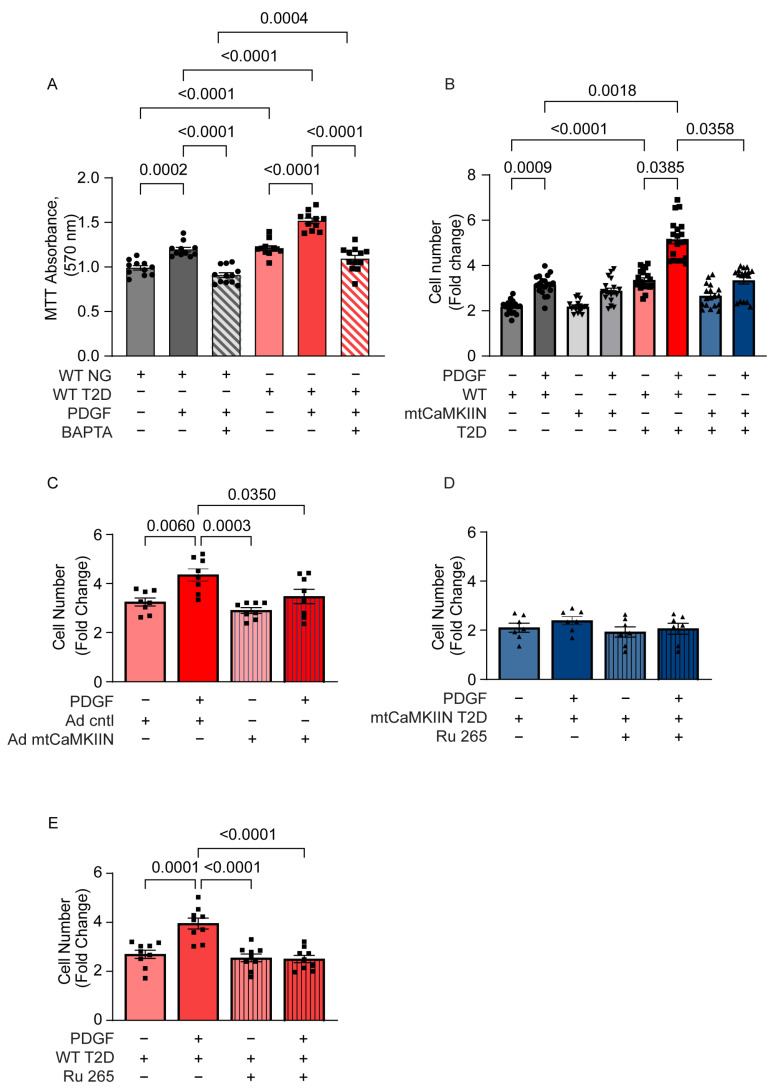
Inhibition of mitochondrial CaMKII reduces proliferation of VSMCs isolated from T2D mice. Numbers of VMSCs isolated from NG or T2D mice of the WT and mtCaMKIIN genetic backgrounds, counted after 72 h in culture with or without PDGF (20 ng/mL). Data are expressed as fold change over levels at 0 h. (**A**) NG or T2D mice of the WT background treated with BAPTA (1 µM), (*n* = 11). (**B**) VSMCs of the WT and mtCaMKIIN genetic backgrounds (*n* = 19). (**C**) VSMCs isolated from T2D mice of the WT background and transduced with control or mtCaMKIIN adenovirus (*n* = 8). (**D**,**E**) VSMCs from T2D mice of the WT and mtCaMKIIN background, with or without administration of the MCU inhibitor Ru 265 (100 µM) (*n* = 7 and 9, respectively). Analyses by Kruskal–Wallis test (**A**–**D**).

**Figure 2 ijms-24-12897-f002:**
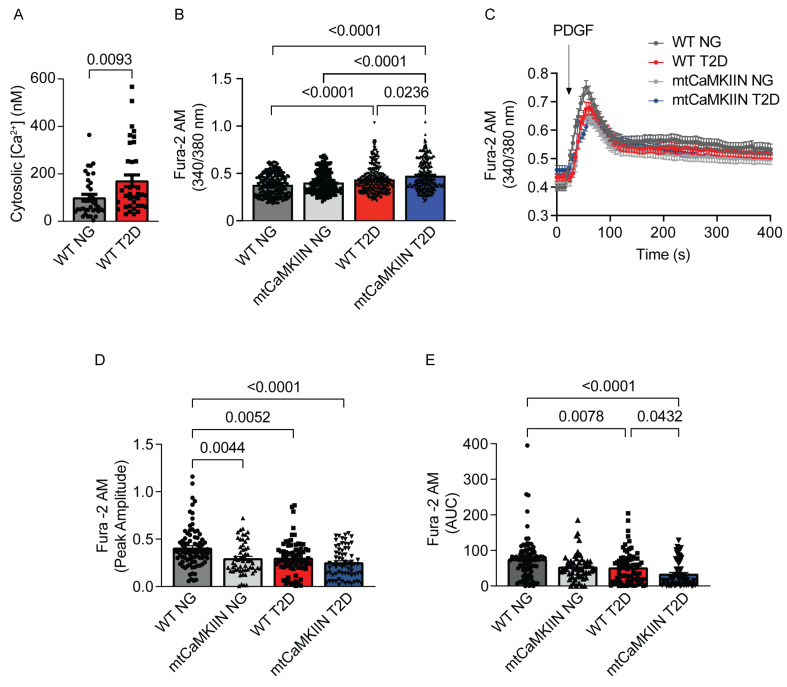
Cytosolic Ca2+ levels and transients are altered by T2D. (**A**) [Ca2+]cyto concentration in VSMCs from WT normoglycemic (NG) and diabetic (T2D) mice (*n* = 6). (**B**) Baseline [Ca2+] levels, as assessed by Fura-2AM imaging, in VSMCs from NG and T2D mice of the WT and mtCaMKIIN genotypes (*n* = 13). (**C**) Cytosolic Ca2+ transients in response to PDGF (*n* = 7). (**D**) Peak amplitude and (**E**) area under the curve (AUC) for Ca2+ transients as in (**C**). Analysis by Mann–Whitney test (**A**), Kruskal–Wallis test (**B**,**D**,**E**).

**Figure 3 ijms-24-12897-f003:**
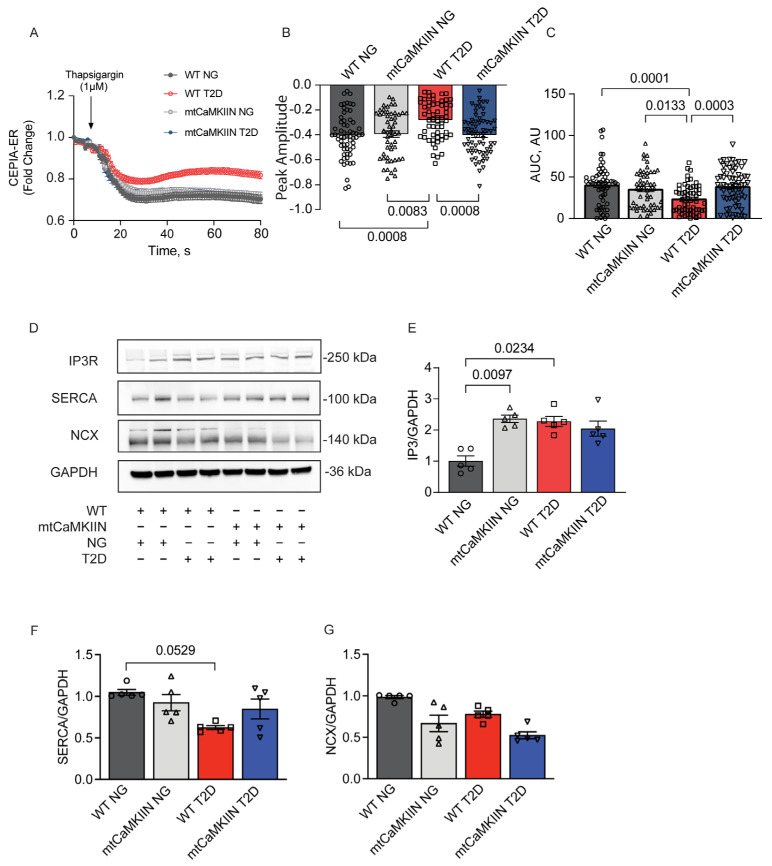
ER Ca2+ release in VSMCs from NG and T2D mice of the WT and mtCaMKIIN genotypes. (**A**) [Ca2+]ER as assessed by CEPIA-ER fluorescence after adding thapsigargin (1 µM) to VSMCs isolated from NG and T2D mice of the WT and mtCaMKIIN genotypes (*n* = 7). (**B**,**C**) Quantitation of (**B**) peak amplitude and (**C**) area under the curve (AUC) for CEPIA-ER fluorescence, from recordings as in (**A**). (**D**) Representative immunoblots for IP3R, SERCA and NCX in whole cell lysates from VSMCs of normoglycemic (NG) and type 2 diabetic (T2D) mice of WT and mtCaMKIIN genotypes. (**E**) Quantification of IP3R adjusted for GAPDH (*n* = 5). (**F**) Quantification of SERCA adjusted for GAPDH (*n* = 5). (**G**) Quantification of NCX adjusted for GAPDH (*n* = 5). Analyses by Kruskal–Wallis test.

**Figure 4 ijms-24-12897-f004:**
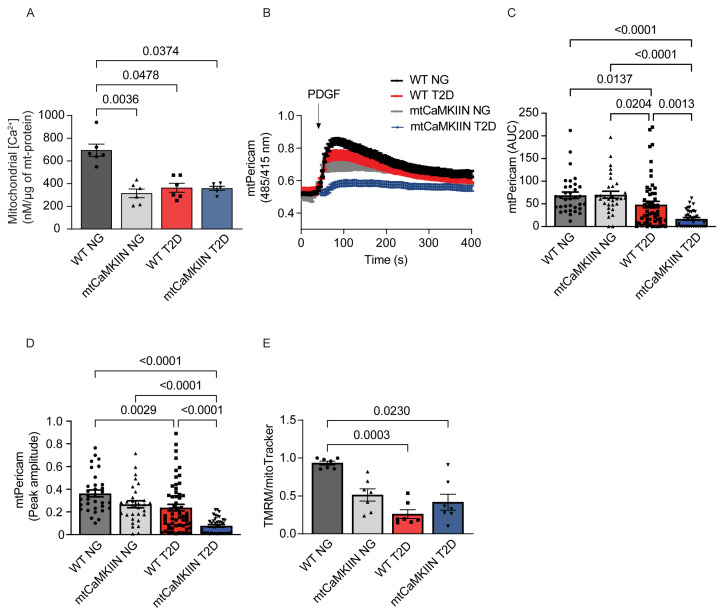
Mitochondrial Ca2+ levels and transients are altered by T2D. (**A**) Matrix-free mitochondrial Ca2+ measured from mitochondria isolated from NG and T2D VSMC of WT and mtCaMKIIN genotypes and normalized to total mitochondrial proteins (*n* = 6). (**B**) Mitochondrial Ca2+ uptake as assessed by mtPericam imaging, in VSMCs from NG and T2D mice of the WT and mtCaMKIIN genotypes induced by PDGF application (20ng/mL) (*n* = 8). (**C**) Quantification of area under the curve (AUC) and (**D**) peak amplitude of mtPericam recordings as in (**B**). (**E**) Mitochondrial membrane potential as assessed by TMRM imaging followed by normalization to the mitoTracker signal in VSMCs from NG and T2D mice of the WT and mtCaMKIIN genotypes (*n* = 7). Analysis by Kruskal–Wallis test.

**Figure 5 ijms-24-12897-f005:**
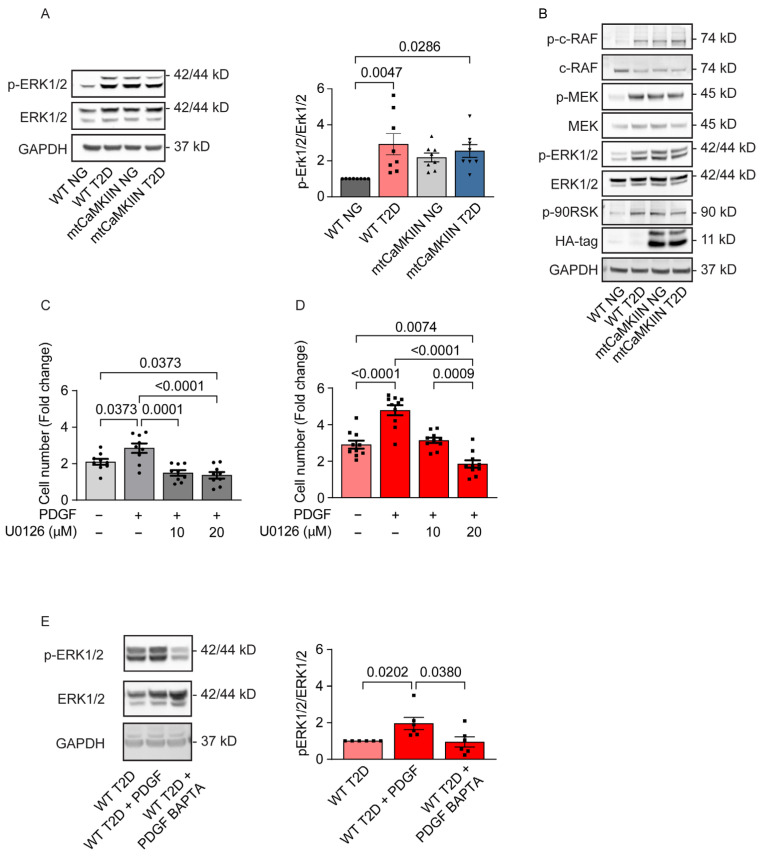
The enhanced proliferation of VSMCs isolated from T2D mice is caused by Ca2+-dependent Erk1/2 proliferation. (**A**) Representative immunoblots for phosphorylated (active) Erk1/2 and total Erk1/2 protein, and their quantification for cultured VSMCs isolated from NG and T2D WT and mtCaMKIIN mice (*n* = 8). (**B**) Representative immunoblots for signaling pathway components upstream of Erk1/2 activation in cultured VSMCs from T2D mice of the WT and mtCaMKIIN genetic backgrounds. (**C**,**D**) Fold changes in cell number in (**C**) WT NG (*n* = 9), (**D**) WT T2D (*n* = 10←) VSMCs 72 h after addition of PDGF and the Erk1/2 inhibitor U0126. (**E**) Representative immunoblot assessing phosphorylated (active) Erk-1/2 and (total) Erk1/2 protein levels and their quantification in cultured cells isolated from T2D mice of the WT genotype 15 min after addition of PDGF and preincubation with the Ca2+ chelator BAPTA (10 µM for 1 h), (*n* = 6). Analyses by Kruskal–Wallis test (**A**,**C**–**E**).

**Figure 6 ijms-24-12897-f006:**
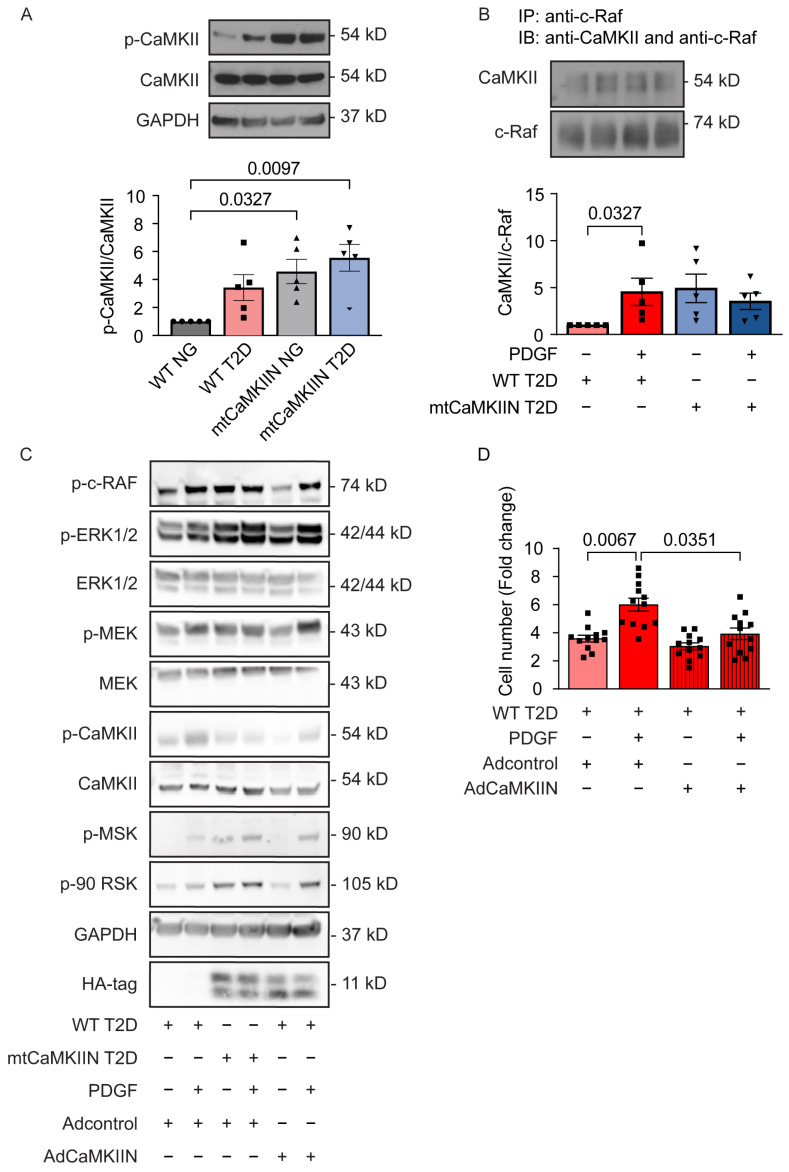
Activation of cytosolic CaMKII in T2D VSMCs mediates Erk1/2 activation. (**A**) Immunoblots for phosphorylated (active) CaMKII and total CaMKII protein in VSMCs isolated from NG and T2D mice of the WT and mtCaMKIIN genotypes (*n* = 5). (**B**) Representative immunoblot (**upper panel**) and quantification (**lower panel**) of co-IP between CaMKII and c-Raf in VSMCs from diabetic (T2D) mice of the WT and mtCaMKIIN genotypes. IP was performed with anti-cRaf and probed with anti-CaMKII (*n* = 5). (**C**) Representative immunoblots assessing upstream signaling components of the Erk1/2 signaling pathway in VSMCs from T2D mice of the WT and mtCaMKIIN genotypes. WT T2D cells were transduced with adenovirus expressing untargeted CaMKIIN for 72 h prior to treatment with PDGF (*n* = 3). (**D**) Number of VSMCs in T2D mice of the WT genotype following transduction with adenovirus expressing untargeted CaMKIIN or control virus for 72 h prior to treatment with PDGF (20 ng/mL) and counted 72 h later (*n* = 12). Analyses by Kruskal–Wallis test (**A**,**B**,**D**).

**Figure 7 ijms-24-12897-f007:**
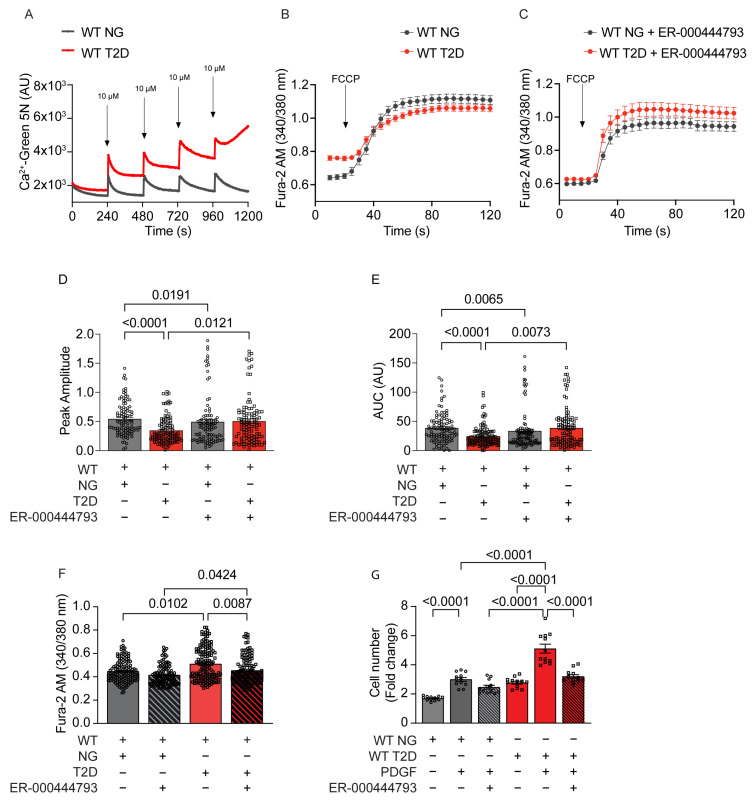
Changes of mitochondrial and cytosolic Ca2+- levels under mPTP inhibition. (**A**) Ca2+ uptake by Calcium Green-5N over time within response to 10 μM CaCl2 in permeabilized VSMCs from WT NG and T2D (*n* = 5). (**B**,**C**) Cytosolic Ca2+ transients induced by FCCP (5 µM) application and measured with Fura-2AM in NG and T2D WT VSMC before and after mPTP inhibitor preincubation (ER-000444793 (10 µM) for 2 h), (*n* = 5). (**D**) Peak amplitude and (**E**) Area under the curve (AUC) quantified from (**B**,**C**). (**F**) Basal Fura-2AM fluorescence measured in NG and T2D WT VSMC with and without mPTP inhibitor preincubation (ER-000444793 (10 µM) for 2 h), (*n* = 5). (**G**) Cell counts at 72 h. after addition of PDGF in NG and T2D WT VSMC before and after mPTP inhibitor preincubation (ER-000444793 (10 µM) for 2 h), (*n* = 12). Analyses by Kruskal–Wallis test (**D**–**G**).

## Data Availability

Not applicable.

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
