# Peer review of "Regulation of Smooth Muscle Cell Proliferation by Mitochondrial Ca2+ in Type 2 Diabetes"

_ijms, 2023, doi:10.3390/ijms241612897_

Round 1

Reviewer 1 Report

This is an animal study, mainly in vitro, regarding smooth muscle cell proliferation by mitochondrial Ca2+ in type 2 diabetes mouse model. The authors concluded that altered Ca2+ handling drives enhanced VSMC proliferation in T2D, with mitochondrial dysfunction contributing to this process.

This reviewer considers that the authors well performed the present study. This reviewer has some comments as described below. 

Major comments:

1.     Figures. Most of them showed the statistically significance, but not big. As the results were a little complicated, the authors should show a graphical abstract to summarize the results. 

2.     Lines 498-506 seem to be unnecessary.

3.     Also, lines 515-542 and lines 546-553 should be correctly described.

4.     The authors should add conclusion at the end of the text. 

Minor comment: 

5.     Ref number in the text was not written normally. The authors should correct them.  

Author Response

We would like to thank the reviewer for the time and effort in evaluating our manuscript. Please find our responses below. All revisions have also been tracked in the manuscript file.

Major comments:

  1. Figures. Most of them showed the statistically significance, but not big. As the results were a little complicated, the authors should show a graphical abstract to summarize the results.
  • We thank the reviewer for this comment. As requested, we added a graphical abstract to display the changes in Ca2+ handling driving enhanced proliferation in T2D as well as the opposing effects of MCU and mPTP inhibition that both decrease smooth muscle cell proliferation in T2D.

  1. Lines 498-506 seem to be unnecessary.
  2. Also, lines 515-542 and lines 546-553 should be correctly described.
  • This text was not part of our submitted manuscript and must have unintendedly been added when typesetting the manuscript. We thank the reviewer for bringing this to our attention. We notified the editorial office with a request to remove the text.

  1. The authors should add conclusion at the end of the text.
  • As requested, we added a section on conclusions at the end of the text.

Minor comment:

  1. Ref number in the text was not written normally. The authors should correct them.
  • We corrected this error.

Reviewer 2 Report

The authors present an original study entitled “Regulation of smooth muscle cell proliferation by mitochondrial Ca2+ in type 2 diabetes”.

The results of the presented study are interesting and novel. The study design and methods are appropriate to the objectives of the study. The conclusion is fully reasoned.

I have a few minor points to address:

1.    “All animal procedures were approved by the University of Iowa Institutional Animal Care and Use Committee”. Please provide the approval protocol number.

2.    Page 16 lines 498-506. Please remove the help comments from the template.

3.    Figure 7G - figure captions need to be centered on the rows.

4.    In the discussion, it would be valuable to briefly discuss the findings in the context of VSMC phenotypic plasticity. Are there associations between impaired calcium metabolism in VSMCs and mitochondrial dysfunction, and VSMC phenotype switching?

Author Response

The authors present an original study entitled “Regulation of smooth muscle cell proliferation by mitochondrial Ca2+ in type 2 diabetes”. The results of the presented study are interesting and novel. The study design and methods are appropriate to the objectives of the study. The conclusion is fully reasoned.

We would like to thank the reviewer for the time and effort in evaluating our manuscript.

I have a few minor points to address:

  1. “All animal procedures were approved by the University of Iowa Institutional Animal Care and Use Committee”. Please provide the approval protocol number.
  • We apologize for this oversight. We added this information in the section on “Materials and Methods”.
  1. Page 16 lines 498-506. Please remove the help comments from the template.
  • This text was not part of our submitted manuscript and must have unintentionally been added when typesetting the manuscript. We thank the reviewer for bringing this to our attention. We notified the editorial office with a request to remove the text.
  1. Figure 7G - figure captions need to be centered on the rows.
  • The figure was edited as requested.
  1. In the discussion, it would be valuable to briefly discuss the findings in the context of VSMC phenotypic plasticity. Are there associations between impaired calcium metabolism in VSMCs and mitochondrial dysfunction, and VSMC phenotype switching?
  • We thank the reviewer for this interesting comment. To our knowledge, there are no data that directly focus on this association. However, the number of mitochondria is elevated in dedifferentiated versus differentiated smooth muscle cells (Fang Y, Pharmacol. 2022). Moreover, several studies have addressed differences in the expression of ion channels in the context of dedifferentiation of smooth muscle cells. Examples include changes in the expression of TRPC and the loss of the CaV1.2 (L-type voltage-gated calcium) channel, which occurs to enable membrane hyperpolarization that increases calcium entry coupled with cell cycle activity, cell proliferation and cell migration (Beech DJ. Biochem Soc Trans, 2007). However, mitochondrial calcium channels were not included in this study. In our hands, the deletion of MCU in normoglycemic VSMCs reduces metabolic activity and blocks cell cycle progression by inhibiting ATP production in G1/S phase (Koval OM, Sci Signal, 2021). This current manuscript does not directly address phenotype switching. We added a comment in this regard in the discussion. Of note, ongoing studies in our lab are currently exploring these aspects.

Reviewer 3 Report

In the article by Koval et al, the authors explore the role of cytoplasmic Ca2+ specifically evaluating the response and proliferation of VSMCs in the pathogenesis of T2D. Overall, the paper offers a thorough exploration of the mechanism of calcium handling in VSMCs and is likely to be of interest to readers 

Comments:

Was this model of STZ-induced T2D described and validated elsewhere? There was no reference in the methods (lines 368-374) or in the introduction. This protocol appears to diverge from more standard models of T2D (e.g. PMID 33905609, 33151911) and further description or reference to detailed model validation would be useful to readers to better interpret results.

Author Response

In the article by Koval et al, the authors explore the role of cytoplasmic Ca2+ specifically evaluating the response and proliferation of VSMCs in the pathogenesis of T2D. Overall, the paper offers a thorough exploration of the mechanism of calcium handling in VSMCs and is likely to be of interest to readers.

Comments:

Was this model of STZ-induced T2D described and validated elsewhere? There was no reference in the methods (lines 368-374) or in the introduction. This protocol appears to diverge from more standard models of T2D (e.g. PMID 33905609, 33151911) and further description or reference to detailed model validation would be useful to readers to better interpret results.

  • We used this model because it mimics some features of human T2D with insulin resistance and mild dysfunction in β-cells without completely compromising insulin secretion.
  • This model was used in rats (Srinivasan K, Pharmacol Res, 2005) and adapted for use in mice (PMID: 22164157, 30891549).
  • Mice were fed a high-fat diet (HFD, 60% fat) for 8 weeks. Then streptozotocin (STZ) (75 mg/kg, 50 mg/kg) was administered by i.p. injections as two separate doses per published protocols (PMID: 22164157, 30891549). Normoglycemic control mice were kept on normal chow. At 12 weeks after initiation of high-fat diet, we confirmed that this model recapitulates T2D phenotypes, including increased body weight up to 40 g, random blood glucose levels above 250 mg/dl, impaired glucose tolerance, hypercholesteremia, and hyperinsulinemia. No significant differences were observed between the genotypes.
  • Description of the T2D model is expanded in the section on “Materials and Methods”, and citations are added to the list of References.

Reviewer 4 Report

To study VSMC functions from a calcium perspective is very interesting. Although I am not entirely sure that the VSMC proliferation is the main issue contributing to cardiovascular pathology in DM2 patients, that is not such a problem, since the concept of changing calcium levels in the different cellular compartments influencing VSMC function is. However, just to show that enhanced glycolysis is a trigger for VSMC proliferation, which may be relevant in early DM2, but less so in late DM2 where medication is needed. J Pathol. 2023 Apr;259(4):388-401. doi: 10.1002/path.6052. Since calcium has so many functions in VSMCs, also contraction/hypertension and mitochondrial function, which influences almost everything, I think the disturbed calcium will influence many different processes in VSMCs and thus change its entire phenotype and thus function, of which proliferation is just one example. So perhaps this could be mentioned somewhere to nuance the role of proliferation in DM2. In line with this, does the disturbed mito calcium directly influence oxidative phosphorylation, as demonstrated by reduced oxygen consumption rate (in a Seahorse assay)? VSMC phenotypic changing is a popular topic these days and could perhaps be mentioned in the discussion, that uncerstanding this calcium handling that causes a phenotypic switch is useful to understand for all vascular diseases. 

It would be nice to make an attractive mechanistic summary as last Figure of the results regarding calcium handling in the various cellular compartments and the important transporters and how they are dysregulated under type 2 DM, since it is quite complex and an overview would help.

Author Response

To study VSMC functions from a calcium perspective is very interesting. Although I am not entirely sure that the VSMC proliferation is the main issue contributing to cardiovascular pathology in DM2 patients, that is not such a problem, since the concept of changing calcium levels in the different cellular compartments influencing VSMC function is. However, just to show that enhanced glycolysis is a trigger for VSMC proliferation, which may be relevant in early DM2, but less so in late DM2 where medication is needed. J Pathol. 2023 Apr;259(4):388-401. doi: 10.1002/path.6052.

  • We thank the reviewer for this interesting comment. This paper was included in the discussion.

Since calcium has so many functions in VSMCs, also contraction/hypertension and mitochondrial function, which influences almost everything, I think the disturbed calcium will influence many different processes in VSMCs and thus change its entire phenotype and thus function, of which proliferation is just one example. So perhaps this could be mentioned somewhere to nuance the role of proliferation in DM2.

  • We thank the reviewer for this interesting comment. We edited the discussion to include these considerations.

In line with this, does the disturbed mito calcium directly influence oxidative phosphorylation, as demonstrated by reduced oxygen consumption rate (in a Seahorse assay)? VSMC phenotypic changing is a popular topic these days and could perhaps be mentioned in the discussion, that understanding this calcium handling that causes a phenotypic switch is useful to understand for all vascular diseases. 

  • We agree that in T2D, glycolysis rather than oxidative phosphorylation is utilized for ATP generation. We also believe that mitochondrial metabolism also supports proliferation by anaplerosis of TCA cycle intermediates to generate building blocks. However, our study does not directly address phenotypic changing. We are currently completing a study that addresses metabolic changes in T2D with respect to VSMC proliferation.
  • We thank the reviewer for the comment in phenotype switching. We included this consideration in the revised discussion.

It would be nice to make an attractive mechanistic summary as last Figure of the results regarding calcium handling in the various cellular compartments and the important transporters and how they are dysregulated under type 2 DM, since it is quite complex, and an overview would help.

  • As requested, we added a graphical abstract.

Round 2

Reviewer 1 Report

Although the authors revised that they add a graphical abstract, this reviewer was not able to find out. Where is it?  

Author Response

Please see in the attachment.

Round 3

Reviewer 1 Report

This graphical abstract is fine, and I can accept the paper.